# Photoinduced DNA Cleavage and Photocytotoxic of Phenanthroline-Based Ligand Ruthenium Compounds

**DOI:** 10.3390/molecules26113471

**Published:** 2021-06-07

**Authors:** Xia Hu, Ning-Yi Liu, Yuan-Qing Deng, Shan Wang, Ting Liu, Xue-Wen Liu

**Affiliations:** 1Hunan Province Cooperative Innovation Center for the Construction & Development of Dongting Lake Ecological Economic Zone, College of Chemistry and Materials Engineering, Hunan University of Arts and Science, Changde 415000, China; xiahu@huas.edu.cn (X.H.); liuningyi@st.huas.edu.cn (N.-Y.L.); Dengyuanqing@st.huas.edu.cn (Y.-Q.D.); Wangshan@st.huas.edu.cn (S.W.); Liuting@st.huas.edu.cn (T.L.); 2Hunan Provincial Key Laboratory of Water Treatment Functional Materials, Hunan University of Arts and Science, Changde 415000, China; 3Hunan Province Engineering Research Center of Electroplating Wastewater Reuse Technology, Hunan University of Arts and Science, Changde 415000, China

**Keywords:** ruthenium complex, photoinduced cleavage, DNA interaction, photocytotoxic

## Abstract

The photophysical and biological properties of two new phenanthroline-based ligand ruthenium complexes were investigated in detail. Their DNA interaction modes were determined to be the intercalation mode using spectra titration and viscosity measurements. Under irradiation, obvious photo-reduced DNA cleavages were observed in the two complexes via singlet oxygen generation. Furthermore, complex **2** showed higher DNA affinity, photocleavage activity, and singlet oxygen quantum yields than complex **1**. The two complexes showed no toxicity towards tumor cells (HeLa, A549, and A375) in the dark. However, obvious photocytotoxicities were observed in the two complexes. Complex **2** exhibited large PIs (phototherapeutic indices) (ca. 400) towards HeLa cells. The study suggests that these complexes may act as DNA intercalators, DNA photocleavers, and photocytotoxic agents.

## 1. Introduction

The DNA-binding behaviors of small molecules have provoked intense interest because DNA has usually been regarded as the most important drug target for anti-tumor activity [1,2,3,4,5,6,7,8]. Many studies have shown that most of the perturbations in cellular processes may come from the different DNA-binding modes of small molecules [8,9,10,11,12]. In general, non-covalent and covalent binding modes have been found in the interactions between small molecules and DNA. For example, well-known platinum-based complexes show their anti-tumor activity by covalent binding to DNA [8,9,10,11,12], which can affect the topological structure of DNA. Meanwhile, many DNA binders have also been found to display good biological activities through non-covalent interactions, especially DNA intercalation. Recently, many intercalators have been reported to display anti-tumor activities because the DNA interactions of intercalators can induce conformational changes in DNA or normal DNA–protein interactions [13,14,15,16,17,18,19,20,21,22,23,24]. Ruthenium complexes have been frequently considered for their possible application as important biological agents due to their strong DNA binding abilities, rich photoactivity and easily constructed coordination geometry [15,16,17,18,19,20,21,22,23,24]. Most of these applications are regarded to originate from the high DNA affinity of ruthenium-based compounds. For example, [Ru(bpy)^2^(dppz)]^2+^ displays a “light switch” effect during its intercalation into DNA [25]. Furthermore, many studies have revealed that changes in the structure of ligands can lead to interesting differences in DNA affinity, reactive oxygen species (ROS) quantum yields, and the DNA photocleavage abilities of ruthenium-based compounds, such as their substituent effect, and the shape and planarity of ligands (main ligands and ancillary ligands). Therefore, a further study is necessary to obtain new intercalative ligands and search for new ruthenium-based compounds with excellent bioactivities. 

On the other hand, photodynamic therapy (PDT) agents under irradiation have shown photocytotoxic activities towards cancer cells via ROS mechanisms. The first PDT drug containing a porphyrin unit, photofrin^®^, has been approved by the Federal Drug Administration (FDA) for the treatment of solid tumors. Recently, ruthenium complexes have been found to activate molecular oxygen into singlet oxygen (^1^O_2_) and to display favorable singlet oxygen quantum yields. This indicates that ruthenium-based compounds have the potential to act as PDT agents. Previous studies show that several kinds of ruthenium-based compounds have been used to confirm their photocytotoxic activities towards tumor cells, including Ru(II) polypyridyl complexes, ruthenium(II)-porphyrin conjugates, and cyclometalated ruthenium(II) complexes [26,27,28,29,30,31,32,33,34,35,36,37,38,39]. These ruthenium complexes show significant phototherapeutic indices due to their high singlet oxygen quantum yields as the result of efficient singlet oxygen photosensitization. Therefore, a further study is necessary to obtain new ruthenium-based compounds with significant singlet oxygen quantum yields and search for new phototherapeutic agents. 

This work stems from our interest in obtaining new ruthenium complexes and investigating their potential biological activities. A phen (phen = 1,10-phenanthroline) unit is usually introduced into organic molecules as a fluorophore to enhance their emission intensities. Furthermore, many ruthenium complexes containing a phen unit display excellent photophysical properties. Furthermore, an aromatic unit (naphthyl) was also introduced to form a naphthoimidazole unit due to its potential photophysical abilities as a fluorophore. In addition, the large aromatic plane of the naphthoimidazole unit is favorable to the DNA binding abilities of ruthenium complexes. We expected good photophysical properties, high ^1^O_2_ quantum yield and excellent bioactivities, all of which can be improved by the modification of the main ligand. Herein, we synthesized two ruthenium complexes containing a new phen-based ligand: 2-(5-(1,10-phenanthroline))-1H-naphtha[2,3]imidazole (pni), [Ru(bpy)_2_(pni)](PF_6_)_2_, and [Ru(phen)_2_(pni)](PF_6_)_2_ (bpy = 2,2′-bipyridine). The DNA interaction, DNA photocleavage abilities, cytotoxicities in the dark, and photocytotoxicities of the two complexes were further studied.

## 2. Results and Discussion

### 2.1. Synthesis and Characterization

The condensation reaction of diamine and aldehyde in the presence of NaHSO_3_ in dimethylacetamide enables the creation of the ligand pni through the formation of an imidazole ring. Precursor Ru complexes and pni were combined in ethylene glycol and refluxed for 8 h to create the desired ruthenium complex (Scheme 1). Structural characterizations were carried out using ESI–MS spectra, NMR spectra, and elemental analysis (Appendix A). Using the MS spectra, the structures of the two complexes were identified by the presence of two peaks, [M-PF_6_]^+^ and [M-2PF_6_]^2+^. ^1^H-NMR spectra showed the proton signals at 13.47 and 13.48 for **1** and **2**, respectively. These peaks can be attributed to the imidazole proton, which confirms the formation of imidazole ring. 

### 2.2. UV-Vis Spectra

UV–visible spectra are used to determine the DNA affinities of ruthenium-based compounds. In general, when the complexes display a high DNA affinity, large decreases in absorbance will occur. Meanwhile, little change will be observed for complexes with a low DNA affinity.

Figure 1 depicts the UV-Vis spectra of two Ru complexes incubating calf thymus DNA (ct-DNA) ([Ru] = 20 μM) in a tris buffer (5 mM tris, 50 mM NaCl, pH = 7.2). The MLCT (metal–ligand charge transfer) bands of complexes **1** and **2** were found to be 452 nm and 445 nm, respectively. With an increase in the amount of DNA, the absorption intensities of **1** and **2** dropped by about 11.5% and 13.6%, respectively. The DNA affinities of the two complexes were evaluated using the DNA binding constant K_b_, which can be calculated from the decrease in the absorbance at the MLCT absorption bands using Equations (1) and (2) [40]. The values of K_b_ are 2.84 ± 0.10 × 10^5^ M^−^^1^ (s = 2.01) and 4.35 ± 0.14 × 10^5^ M^−^^1^ (s = 3.39) for **1** and **2**, respectively. [Ru(bpy)_2_(dppz)]^2+^ has been reported to be a strong DNA binder through its intercalation with K_b_ of 4.9 × 10^6^ M^−^^1^ [41,42]. Comparing the K_b_ of our complexes to those of classical intercalators, our complexes displayed lower DNA affinities. The difference in DNA affinities between our complexes and those classical intercalators may be caused by the plane area of intercalative ligand. Furthermore, **2** presented a higher DNA affinity than **1**. This is likely to be due to the effect of the ancillary ligand. As previous reports have shown, ancillary ligands possess a large aromatic planarity area and high hydrophobicity, usually leading to the high DNA affinity of Ru complexes [43].

### 2.3. Viscosity Measurements

Another useful method to determine DNA binding modes is a viscosity experiment, which is usually used to measure changes in the length of DNA when small molecules interact with DNA. When a probe interacts with DNA, different modes can affect DNA viscosity to different degrees. For example, ethidium bromide (EB) can cause DNA viscosity to increase because it can bind with DNA by intercalation, which causes an increase in DNA length. Conversely, [Ru(bpy)_3_]^2+^ has little effect on DNA viscosity due to its electrostatic binding mode with DNA [44,45,46]. Therefore, changes in DNA viscosity can enable the determination of DNA binding modes. The changes in DNA viscosities for EB, [Ru(bpy)_3_]^2+^, **1** and **2** are shown in Figure 2. From Figure 2, we can see that the two complexes had a similar impact on DNA viscosity to that of EB, indicating that complex **1** and **2** interact with DNA by intercalation. Lower DNA affinities were observed for **1** and **2** compared to EB because the DNA binding mode and DNA affinity are important factors that lead to the change in the DNA viscosity. Additionally, complex **2** presents stronger DNA interaction than complex **1** based on the viscosity experiment results.

### 2.4. Emission Titration Experiments

The good photophysical properties of ruthenium-based complexes can be utilized to study their DNA interaction, because the addition of DNA can cause the perturbation of the emission properties of ruthenium complexes. The luminescence behaviors of the complexes were tested by adding ct-DNA to the solutions of the complexes, which is helpful to further understand the DNA affinity of these compounds. The results can be seen in Figure 3. When excited at 450 nm, **1** and **2** emitted red luminescence at 606 nm and 600 nm, respectively. After the complexes interacted with DNA, a large increase in emissions was observed, and the increasing ratio of emission intensity was ca. 2.34- and 1.75-times for complex **1** and **2**, respectively. Here, DNA caused the large perturbation of the emission intensity of the two complexes, which demonstrated that the two complexes exhibited a high DNA affinity. 

[Fe(CN)_6_]^4−^ can repulse DNA since both of them are polyanionic species. As previously reported, ferrocyanide can quench the luminescence of the complexes. When DNA interacts with the complexes, the emission quenching will occur due to the electrostatic attraction and coulombic repulsion between ferrocyanide and DNA. The results are shown in Figure 4. Here, similar emission-quenching behaviors were observed for our complexes. Furthermore, a higher quenching efficiency was observed in complex **1** due to its lower DNA-binding ability, as compared to complex **2**. However, Turro has reported that weak luminescence quenching cannot prove the presence of intercalation, because weak luminescence quenching has also been observed for some complexes bound through electrostatic interaction [47]. Therefore, luminescence quenching by ferrocyanide can only reflect the degree of the DNA affinity of a complex; the binding mode cannot be determined using luminescence quenching data. These results also demonstrate that complex **2** exhibited stronger DNA interaction than **1**.

### 2.5. DNA Photocleavage Studies

Ruthenium-based compounds are well known for their good photophysical properties and can serve as photoinduced metallonucleases. The main reason for this is that such compounds can activate molecular oxygen under irradiation, form reactive oxygen species, and cleave DNA [34,48]. The photo-reduced nuclease activities of the two complexes in this experiment were tested using agarose gel electrophoresis. Figure 5 depicts the photo-reduced DNA cleavage results for complex **1** and **2**. The amount of Form I (the intact supercoil form) decreased and the amount of Form II (the nicked circular form) increased. Many polypyridyl ruthenium complexes have been reported to be DNA photocleavers [30,43,49,50]. For example, [Ru(bpy)_2_dppn]^2+^ (dppn = 4,5,9,16-tetraaza-dibenzo[a,c]naphthacene) can completely cleave Form I into Form II at 5 μM [49]. Meanwhile, [Ru(dmb)_2_(NMIP)]^2+^ cleaves most of Form I at 60 μM [50] (NMIP = 2′-(2′′-nitro-3′′,4′′-methylenedioxyphenyl)imidazo-[4′,5′-*f*][1,10]-phenanthroline, dmb = 4,4′-dimethyl-2,2′-bipyridine). Although our complexes displayed lower photocleavage abilities than that of [Ru(bpy)_2_dppn]^2+^, obvious DNA-cleaving activities were observed for the two complexes, which demonstrated that the complexes were potentially light-activated compounds. Furthermore, **2** displayed a stronger cleaving ability compared to **1**.

The possible mechanism of photo-reduced cleavage was investigated by determining the presence of reactive oxygen species. Different scavengers were added into the photocleavage system under irradiation, such as mannitol, histidine, DMSO (DMSO = dimethyl sulfoxide), sodium azide, and SOD (SOD = superoxide dismutase) [34,43,48]. These ROS scavengers can quench different reactive oxygen species and affect the DNA-cleaving ability of a complex. As seen in Figure 6, the obvious inhibition of DNA cleavages was observed after the addition of NaN_3_ and histidine to the two complexes. The results showed that ^1^O_2_ (singlet oxygen) appeared in the DNA cleavage system, indicating that the two complexes can produce singlet oxygen under irradiation. However, no obvious inhibition of photocleavage activity was caused by SOD, DMSO or mannitol, indicating that OH• and O_2_•^−^ were not formed by the two complexes under irradiation. This suggested that DNA photo-reduced cleavage should be initiated by ^1^O_2_ for complexes **1** and **2**. 

To further confirm the presence of ^1^O_2_ in the DNA photocleavage, DPBF was used to measure the quantum yields of ^1^O_2_. DPBF is a typical inhibitor of singlet oxygen and can emit strong fluorescence at 479 nm. When it reacts with singlet oxygen, the amount of DPBF will decrease, leading to fluorescence quenching. Under irradiation at 450 nm, the fluorescence quenching of DPBF by the two complexes was observed, further confirming the presence of singlet oxygen (Figure 7). Compared to [Ru(bpy)_3_]^2+^ ($\Phi_{\Delta }^{S}$ = 0.81 [51]), the $\Phi_{\Delta }$ values of complexes **1** and **2** were 0.79 and 0.82 according to Equation (S1) and (S2), respectively. The results indicated that they can produce ^1^O_2_. Combined with the photocleavage results, ^1^O_2_ is further confirmed to be the main reactive species during the DNA photocleavage process.

### 2.6. Photocytotoxicity

The MTT method was used to assess the photocytotoxicities and dark cytotoxicities in vitro for the two complexes against HeLa, A549, and A375 cells. The IC50 values of complex **1**, complex **2**, and cisplatin are given in Table 1. Cisplatin showed obvious dark cytotoxicities against all cancer cells and no significant photocytotoxicities were observed. After 10 min irradiation, the two ruthenium complexes displayed significant photocytotoxicities towards all cancer cells. The photocytotoxicity index (PI) value of complex **1** against HeLa, A549, and A375 cells was 208, 227, and 244, respectively. For complex **2**, the PI value was 400, 357, and 384, respectively. The cell viabilities of A549 cells in the presence of complexes **1** and **2** are shown in Figure 8. The two complexes exhibited significant inhibitory effects on cell viability (for HeLa and A375 cells (Appendix A), see Appendix A). However, the two complexes did not cause obvious cytotoxicities in the dark against all cancer cells (>100 μM). The results showed that the two complexes displayed low dark cytotoxicity and high phototoxicity, indicating that the two complexes can act as a potential PDT candidate. Complex **2** displayed higher photocytotoxicity than complex **1**, since complex **2** displayed higher singlet oxygen quantum yields compared to complex **1**. Ruthenium complexes containing phen as a co-ligand usually exhibit higher singlet oxygen quantum yields than complexes with the co-ligand bpy, due to the difference in the rigidity of the ancillary ligand. Previous studies have shown that their parent complex—[Ru(bpy)_3_]^2+^ [32]—exhibited low cytotoxicity toward HeLa and A549 cells under light irradiation, although it displays a high ^1^O_2_ quantum yield, indicating that ^1^O_2_ quantum yield is not the only factor affecting photocytotoxocity. However, its derivative complex, [Ru(bpy)_2_dppn]^2+^ [52], displayed high photocytotoxocity after the modification of the main ligand. Therefore, these factors affected photocytotoxocity. 

## 3. Materials and Methods

### 3.1. Instrumentation

ESI mass spectra were obtained using an LCQ system (Finnigan MAT, San Jose, CA, USA). Microanalysis was performed using a Perkin–Elmer 240Q elemental analyzer. ^1^H-NMR and ^13^C-NMR spectra were collected in (CD_3_)_2_SO using a Bruker 2000 spectrometer. 

### 3.2. DNA Interactions

The DNA interactions of the complexes were assessed using UV titration at room temperature. The concentration of the ct-DNA solution was determined by absorption spectroscopy [49,50,53]. After the addition of ct-DNA into the solutions of the ruthenium complexes (20 μM), UV-Vis spectra were recorded every 5 min. McGhee’s equations (Equations (1) and (2)) were used to determine the values of *K* for the ruthenium-based compounds **1** and **2** [40].
(*ε_a_* − *ε_f_*)/(*ε_b_* − *ε_f_* ) = (*b* − (*b*^2^−2*K*^2^*C_t_*[DNA]/*s*)^1/2^)/2*KC_t_*(1)
*b* = 1 + *KC_t_* + *K*[DNA]/2*s*(2)
where *ε_a_*, *ε_f_*, and *ε_b_* corresponds to the molar absorptivity of the DNA-bound complex, the free complex, and the DNA-saturated complex, respectively. *s* is the binding site size. Here, the value of *C_t_* is 20 μM for the ruthenium complexes. *K* is the binding constant of the ruthenium complexes. 

Emission titration experiments were performed by adding various concentrations of ct-DNA into the solutions of the ruthenium complexes (5 μM) at room temperature. Emission quenching experiments were also carried out, using [Fe(CN)_6_]^4−^ as the quencher. In general, various concentrations [Fe(CN)_6_]^4−^ were added to the solutions of the ruthenium complexes (5 μM) or Ru-DNA ([Ru] = 5 μM; [DNA] = 400 μM). The luminescence spectra were recorded in the range of 500 to 800 nm with an excitation wavelength of 450 nm.

### 3.3. Photoinduced DNA Cleavage

The photoinduced cleavage experiments were carried out under irradiation at 365 nm. Various concentrations of the Ru compounds were added to solutions of pBR322 DNA (0.1 μg). Then, the mixture was irradiated at 365 nm. Gel electrophoresis was used to evaluate the DNA cleaving abilities of the complexes under irradiation. 

### 3.4. Singlet Oxygen Quantum Yield Measurement

The quantum yields (ΦΔ) of ^1^O_2_ were tested using an ^1^O_2_ quencher and DPBF (1,3-diphenylisobenzofuran) in CH_3_OH. The methanol solutions of DPBF and the complexes were irradiated at 450 nm. The fluorescence spectra of the DPBF were collected every 3 min (λ_ex_ = 405 nm, λ_em_ = 479 nm). Quantum yields (ΦΔ) of ^1^O_2_ were obtained according to the reported equation [53] (Appendix A).

### 3.5. Photocytotoxicity

Hela, A549, and A375 cells (5000 cells) were cultured separately in 96-well plates overnight. Various concentrations of the Ru compounds or cisplatin were added to the cells and kept in the dark for 12 h. A fresh medium was used for subsequent experiments. The cells were irradiated for 10 min (LED system 450 nm, 6 mW/cm^2^) and kept in the dark for another 36 h. The standard MTT method was used to obtain the IC50 values of the complexes.

### 3.6. Synthesis

2,3-diaminonaphthalene, was obtained commercially. 1,10-phenanthroline-5-carbaldehyde, and the precursor ruthenium complexes, [Ru(L)_2_Cl_2_].2H_2_O (L = bpy and phen), were prepared according to the reported method [54,55,56]. 

#### 3.6.1. 2-(5-(1,10-Phenanthroline))-1H-naphtha[2,3]imidazole (pni)

NaHSO_3_ (0.208 g, 2.0 mmol), 2,3-diaminonaphthalene (0.158 g, 1.0 mmol), and 1, 10-phenanthroline-5-carbaldehyde (0.208 g, 1.0 mmol) were added to 10 mL dimethylacetamide. The solution was refluxed for 6 h. Then, 150 mL of water was added, yielding a dark brown precipitate. Yield: 81.1%. Anal (%): ESIMS: *m/z* = 347 ([M + 1] +). ^1^H-NMR (400 MHz, ppm, DMSO-d^6^): 9.83 (d, 1H, *J* = 8.0 Hz), 9.21 (d, 2H, *J* = 4.0 Hz), 8.71 (s, 1H), 8.63 (d, 1H, *J* = 8.0 Hz), 8.26 (s, 2H), 8.08 (dd, 2H, *J*_1_ = 4.0 Hz, *J*_2_ = 4.0 Hz), 7.93 (dd, 1H, *J*_1_ = 4.0 Hz, *J*_2_ = 4.0 Hz), 7.88 (dd, 1H, *J*_1_ = 4.0 Hz, *J*_2_ = 4.0 Hz), 7.43 (dd, 2H, *J*_1_ = 4.0 Hz, *J*_2_ = 4.0 Hz).

#### 3.6.2. Synthesis of Complexes **1** and **2**

The pni (0.140 g, 0.4 mmol) was mixed with an ethylene glycol (10 mL) solution of the precursor ruthenium complexes (0.4 mmol). After refluxing under argon for 8 h, the addition of KPF_6_ yielded a dark red precipitate. The final products were obtained using an aluminium oxide column with 20% toluene in acetonitrile. 

[Ru(bpy)_2_(pni)](PF_6_)_2_ (1), Yield: 40.1%. (Found: C, 49.04; H, 2.90; N, 10.72%. Calcd for C_43_H_30_N_8_F_12_P_2_Ru: C, 49.20; H, 2.88; N, 10.67%). ES-MS (CH_3_CN): *m/z* = 904.6 ([M-PF6]^+^), 379.8 ([M-2PF_6_]^2+^). ^1^H NMR (400 MHz, ppm, DMSO-d^6^): 13.47 (s, 1H), 10.11 (d, 1H, *J* = 8.0 Hz), 9.10 (s, 1H), 8.87 (d, 5H, *J* = 8.0 Hz), 8.41 (s, 1H), 8.24 (m, 4H), 8.18 (s, 1H), 8.12 (d, 4H, *J* = 8.0 Hz), 8.04 (dd, 1H, *J*_1_ = 8.0 Hz, *J*_2_ = 8.0 Hz), 7.97 (dd, 1H, *J*_1_ = 8.0 Hz, *J*_2_ = 8.0 Hz), 7.86 (d, 2H, *J* = 4.0 Hz), 7.67 (dd, 2H, *J*_1_ = 8.0 Hz, *J*_2_ = 8.0 Hz), 7.60 (t, 2H, *J*_1_ = 4.0 Hz, *J*_2_ = 4.0 Hz), 7.47 (t, 2H, *J*_1_ = 8.0 Hz, *J*_2_ = 8.0 Hz), 7.38 (t, 2H, *J*_1_ = 8.0 Hz, *J*_2_ = 8.0 Hz). ^13^C NMR (101 MHz, ppm, DMSO-d^6^): 157.26, 157.03, 153.83, 153.40, 153.10, 151.96, 148.02, 147.85, 144.60, 138.49, 138.37, 137.68, 136.95, 135.33, 131.29, 130.41, 129.87, 129.23, 128.74, 128.36, 128.27, 128.01, 127.58, 127.17, 124.93, 123.92, 116.72, 107.63.

[Ru(phen)_2_(pni)](PF_6_)_2_ (2), Yield: 51.3%. Anal (%): (Found: C, 51.37; H, 2.79; N, 10.27%, Calc for C_47_H_30_N_8_F_12_P_2_Ru: C, 51.42; H, 2.75; N, 10.21%). ES-MS (CH_3_CN): *m/z* = 952.7 ([M-PF6]^+^), 403.8 ([M-2PF_6_]^2+^). ^1^H-NMR (400 MHz, ppm, DMSO-d^6^): 13.48 (s, 1H), 10.11 (d, 1H, *J* = 8.0 Hz), 9.07 (s, 1H), 8.85 (d, 1H, *J* = 4.0 Hz), 8.80 (d, 4H, *J* = 8.0 Hz), 8.41 (s, 4H), 8.21 (m, 4H), 8.11 (m, 4H), 7.93 (dd, 1H, *J*_1_ = 4.0 Hz, *J*_2_ = 4.0 Hz), 7.81 (m, 5H), 7.45 (t, 2H, *J*_1_ = 8.0 Hz, *J*_2_ = 8.0 Hz), 7.25 (t, 1H, *J*_1_ = 8.0 Hz, *J*_2_ = 8.0 Hz), 7.16 (d, 1H, *J*_1_ = 8.0 Hz, *J*_2_ = 8.0 Hz). ^13^C-NMR (101 MHz, ppm, DMSO-d^6^): 154.36, 153.66, 153.47, 153.23, 148.44, 148.29, 147.69, 147.63, 144.64, 137.82, 137.62, 137.37, 136.92, 135.35, 131.30, 130.98, 130.95, 130.40, 129.84, 129.37, 129.21, 128.75, 128.68, 128.54, 128.01, 127.57, 127.46, 127.06, 126.82, 125.79, 124.90, 123.91, 116.72, 107.63.

## 4. Conclusions

In this work, two ruthenium complexes were prepared. The ligand pni was obtained by introducing a naphthoimidazole unit at the 5-position of the phen. These complexes exhibited high DNA affinities through their DNA intercalative mode. Photoinduced DNA cleavages demonstrated that they can cleave DNA effectively. Mechanism experimental results indicated that they can produce singlet oxygen under irradiation and display high ^1^O_2_ quantum yields. Photocytotoxicity results showed that the two complexes displayed low dark cytotoxicity, high phototoxicity, and large PI values. They display great potential for application as photocytotoxic agents through the modification of the main ligand.

## Data Availability

Not applicable.

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
