# Peer review of "Photoinduced DNA Cleavage and Photocytotoxic of Phenanthroline-Based Ligand Ruthenium Compounds"

_molecules, 2021, doi:10.3390/molecules26113471_

Round 1

Reviewer 1 Report

In the manuscript, the authors synthesized new ruthenium complexes with a phenanthroline ligand substituted by a naphthoimidazole unit for photo-induced DNA cleavage. These complexes bound with DNA by intercalation and exhibited the ability of the DNA photocleavage by singlet oxygen generation. In addition, these compounds showed low dark cytotoxicity and high phototoxicity. These results indicate that these ruthenium complexes can be applied potentially to photodynamic therapy.

The manuscript can be accepted after the following points are addressed.

  1. Why did the authors choose the naphthoimidazole unit? What are the effects of this unit on the interactions with DNA, photo-cleavage of DNA and photocytotoxicity?
  2. The authors compared the photocytotoxicity of their compounds with cisplatin. How about the other ruthenium complexes reported previously?
  3. In the Title, “phenthroline” is an incorrect spelling of “phenanthroline”.
  4. On page 5, line 155, “argose gel” is an incorrect spelling of “agarose gel”.

Author Response

Reviewer: 1

Comments and Suggestions for Authors

In the manuscript, the authors synthesized new ruthenium complexes with a phenanthroline ligand substituted by a naphthoimidazole unit for photo-induced DNA cleavage. These complexes bound with DNA by intercalation and exhibited the ability of the DNA photocleavage by singlet oxygen generation. In addition, these compounds showed low dark cytotoxicity and high phototoxicity. These results indicate that these ruthenium complexes can be applied potentially to photodynamic therapy.

The manuscript can be accepted after the following points are addressed.

Why did the authors choose the naphthoimidazole unit? What are the effects of this unit on the interactions with DNA, photo-cleavage of DNA and photocytotoxicity?

Response and revision: The purpose of introducing naphthyl unit has been presented in the manuscript. And the effects of this unit on the properties of ruthenium complexes have also given in the manuscript.

The authors compared the photocytotoxicity of their compounds with cisplatin. How about the other ruthenium complexes reported previously?

Response and revision: The photocytotoxicity of two synthesized compounds have also compared to some ruthenium complexes reported previously. And the related reference has also been cited.

In the Title, “phenthroline” is an incorrect spelling of “phenanthroline”.

On page 5, line 155, “argose gel” is an incorrect spelling of “agarose gel”.

Response and revision: The errors have been corrected.

Reviewer 2 Report

In this manuscript, Xia Hua et al., have synthesized new type of ruthenium complex consisting of 2-(5-(1,10-phenanthroline))-1H-naphtha[2,3]imidazole (pni). The authors demonstrated interaction of the synthesized ruthenium complex with DNA, photo-induced cleavage of DNA and inhibition of cancer cell growth via singlet oxygen produced by irradiation of the complex. The concept of photodynamic therapy using the ruthenium complex is already demonstrated by previous studies, which are shown in references in the introduction. The main purpose of this manuscript seems to be investigation of the physical properties and biological activities of ruthenium complexes with chemical modifications different from those used in the previous studies. Although the authors are discussing their results with comparing the properties of [Ru(bpy)3]2+, as a standard ruthenium complex, the discussion is just comparing the values in reference except Figure 2. In addition, the discussions seem not sufficient to clearly show the physicochemical properties and biological activities of the synthesized complexes based on the unique chemical structures. This reviewer thinks that the authors should scientifically describe the reason that that the authors used pni unit for the modification of ruthenium complex, and what kinds of effect did the pni unit provide to the complex. Additionally, this manuscript is insufficient in its scientific descriptions as this reviewer pointed bellow.

  1. Correct names should be described before the first description of abbreviation.

There are several abbreviations such as ROS, PTD, CT-DNA, FDA and MLCT that do not have correct name at the place first described.

  1. What kind of buffer solution did the author used through in vitro experiments.

There is no information about the buffer conditions.

  1. The figure captions are not enough to catch the experimental conditions and procedures.

Explanations of sample conditions should be described. The designations of X- and Y-axis is also required to catch the analyses process of the experimental data. Especially, the experimental processes shown in Figures 2, 3, and 4 are not described even in the “Materials and Methods”.

  1. What does CT-DNA concentration mean?

CT-DNA is calf thymus DNA, which is long DNA. Does the concentration mean that of phosphate unit? In that case, what is the physical meaning of binding constant (Kb)? If the authors just use the Kb value for comparison with previous studies, the authors should describe that the Kb value is just observed binding constant based on the concentration of phosphate unit in CT-DNA.

  1. Why did the absorbance at 260 nm decreased with titration of CT-DNA in Figure 1b?

DNA itself has absorption at UV range, in which maximum absorbance is at 260 nm. Thus, addition of CT-DNA into sample should increase the absorbance at 260 nm.

  1. What do “Form I” and “Form II” mean in Figures 5 and 6?

This reviewer guesses the Form I is plasmid DNA in supper coil, and Form II is that in open circle state after nicking by photo-induced cleavage of one strand. In that case, the cleavage efficiency seems not so high because the authors should observe shorter cleaved DNA if the ruthenium complex cleaved DNA frequently at close positions. The authors should discuss about the efficiency of DNA cleavage at least comparing to previous studies.

  1. Results of photo-induced prohibition of cell growth or induction of cell death are better to be shown not only in Table 1 but also in additional figure.

The results are important that demonstrated biological activities of the synthesized complexes. Showing decrease in MTT assay signals depending on the complex concentration is recommended. But, as this reviewer suggested, comparing with control ruthenium complexes previously demonstrated is required.

  1. Citations should be carefully provided in text.

There are several sentences with no citations even the sentences describe studies in previous.

Author Response

Reviewer: 2

Comments and Suggestions for Authors

In this manuscript, Xia Hu et al., have synthesized new type of ruthenium complex consisting of 2-(5-(1,10-phenanthroline))-1H-naphtha[2,3]imidazole (pni). The authors demonstrated interaction of the synthesized ruthenium complex with DNA, photo-induced cleavage of DNA and inhibition of cancer cell growth via singlet oxygen produced by irradiation of the complex. The concept of photodynamic therapy using the ruthenium complex is already demonstrated by previous studies, which are shown in references in the introduction. The main purpose of this manuscript seems to be investigation of the physical properties and biological activities of ruthenium complexes with chemical modifications different from those used in the previous studies. Although the authors are discussing their results with comparing the properties of [Ru(bpy)3]2+, as a standard ruthenium complex, the discussion is just comparing the values in reference except Figure 2. In addition, the discussions seem not sufficient to clearly show the physicochemical properties and biological activities of the synthesized complexes based on the unique chemical structures. This reviewer thinks that the authors should scientifically describe the reason that that the authors used pni unit for the modification of ruthenium complex, and what kinds of effect did the pni unit provide to the complex.

Response and revision: The photocytotoxicity of two synthesized compounds have also compared to some ruthenium complexes reported previously. The related paragraphs have been revised.

Additionally, this manuscript is insufficient in its scientific descriptions as this reviewer pointed bellow.

  1. Correct names should be described before the first description of abbreviation.There are several abbreviations such as ROS, PTD, CT-DNA, FDA and MLCT that do not have correct name at the place first described.

Response and revision: The errors have been corrected.

  1. What kind of buffer solution did the author used through in vitro experiments.

There is no information about the buffer conditions.

Response and revision: The buffer haven given in the manuscript.

3.The figure captions are not enough to catch the experimental conditions and procedures.

Response and revision: The figure captions have been revised.

4.Explanations of sample conditions should be described. The designations of X- and Y-axis is also required to catch the analyses process of the experimental data. Especially, the experimental processes shown in Figures 2, 3, and 4 are not described even in the “Materials and Methods”.

Response and revision: The experimental conditions are given in the manuscript.

  1. What does CT-DNA concentration mean?CT-DNA is calf thymus DNA, which is long DNA. Does the concentration mean that of phosphate unit? In that case, what is the physical meaning of binding constant (Kb)? If the authors just use the Kb value for comparison with previous studies, the authors should describe that the Kb value is just observed binding constant based on the concentration of phosphate unit in CT-DNA.
    Response and revision: The concentration of DNA was determined spectrophotometrically. The extinction coefficient for calf thymus DNA is 6600 per nucleotide. Kb is the microscopic equilibrium constant for the binding site composed of base pairs, which is used to evaluate DNA affinity of ruthenium complex.
  2. Why did the absorbance at 260 nm decreased with titration of CT-DNA in Figure 1b?DNA itself has absorption at UV range, in which maximum absorbance is at 260 nm. Thus, addition of CT-DNA into sample should increase the absorbance at 260 nm.

Response and revision: In order to eliminate the absorbance of DNA in the range of 200-400nm, we added the same amount of CT-DNA into the reference cell. Then, the UV-vis spectra can reflect the true response of the interaction between ruthenium complexes and CT-DNA.

  1. What do “Form I” and “Form II” mean in Figures 5 and 6?This reviewer guesses the Form I is plasmid DNA in supper coil, and Form II is that in open circle state after nicking by photo-induced cleavage of one strand. In that case, the cleavage efficiency seems not so high because the authors should observe shorter cleaved DNA if the ruthenium complex cleaved DNA frequently at close positions. The authors should discuss about the efficiency of DNA cleavage at least comparing to previous studies.

Response and revision: Form I and Form II have been defined in the manuscript. The efficiency of DNA cleavage has been discussed by comparing to previous studies. In theory, the linear form of DNA (Form III) may be obtained under the photocleavage experimental conditions. However, from the literature and our experiment results, the linear form of DNA (Form III) have rarely been observed.

  1. Results of photo-induced prohibition of cell growth or induction of cell death are better to be shown not only in Table 1 but also in additional figure.The results are important that demonstrated biological activities of the synthesized complexes. Showing decrease in MTT assay signals depending on the complex concentration is recommended. But, as this reviewer suggested, comparing with control ruthenium complexes previously demonstrated is required.

Response and revision: The photocytotoxicity of two synthesized compounds have also compared to some ruthenium complexes reported previously. And The cell viabilities data have been given in the manuscript.

  1. Citations should be carefully provided in text.

Response and revision: The errors have been corrected.

  1. There are several sentences with no citations even the sentences describe studies in previous.

Response and revision: The errors have been corrected.

Reviewer 3 Report

The Authors present two phenanthroline-based ligand ruthenium complex. They investigate complexes interaction with DNA and study DNA cleavage under irradiation.

In general, the topic of the manuscript is interesting however I have some concerns:

1) First of all, reference to other studies are not always present. More precise reference to other studies should be added, by properly including the references.

2) Authors should always use the entire name before abbreviations.

3) Overall, it is not clear to me the conclusions Authors reach. They carefully describe the experiments, however it is not clear to me how crucial could be the employment of such complexes for tumor applications. A more carefully description of the hypotheses based on the obtained results, their limits and how these can be possibly improved, is in my opinion necessary.

4) Typos and English inaccuray are often present.

Author Response

Reviewer 3

The Authors present two phenanthroline-based ligand ruthenium complex. They investigate complexes interaction with DNA and study DNA cleavage under irradiation.

In general, the topic of the manuscript is interesting however I have some concerns:

1) First of all, reference to other studies are not always present. More precise reference to other studies should be added, by properly including the references.

Response and revision: The references have been updated.

  • Authors should always use the entire name before abbreviations.

   Response and revision: The errors have been corrected.

3) Overall, it is not clear to me the conclusions Authors reach. They carefully describe the experiments, however it is not clear to me how crucial could be the employment of such complexes for tumor applications. A more carefully description of the hypotheses based on the obtained results, their limits and how these can be possibly improved, is in my opinion necessary.

 Response and revision:The related discussion has been improved.

4) Typos and English inaccuray are often present.

 Response and revision: The errors have been corrected.

Round 2

Reviewer 1 Report

All points have been addressed. I think this manuscript is acceptable for publication in Molecules in the current form.

Author Response

Thanks the reviewer 1 for the comment.

Reviewer 2 Report

Authors improved the manuscript according to the previous reviewers' comments.

Now this reviewer thinks the manuscript acceptable after some minor revision.

1. Temperature conditions should be given.

The authors provided experimental buffer conditions in the text and captions. But there are not information regarding to the experimental temperature.

This reviewer still think the "Materials and Methods" section is not enough for readers to follow the experimental procedure.

2. The supporting information seems the same with original version.

The authors mention that the added results of cell viability assay in supporting information (Page 7, line 218). But, the results are missing.

3. Caption of Figure 5 seems wrong.

This reviewer speculates that the Lane 0 is DNA+Ru(20uM) without light irradiation and Lane 1 is DNA only (without addition of Ru) with light irradiation. 

4. There are typos in text.

For example,

"nicked nicked circularsincreased" (Page 5, line 166)

"than that of than" (Page 6, line 173)

This reviewer recommends careful check of their English.

5, Some "1" in "compound 1" are not bolded in paragraph of "2.6 Photocytotoxicity".

6. Line break should be added before "3.3 Photoinduced DNA cleavage".

Author Response

  1. Temperature conditions should be given.

The authors provided experimental buffer conditions in the text and captions. But there are not information regarding to the experimental temperature.

This reviewer still think the "Materials and Methods" section is not enough for readers to follow the experimental procedure.

Response and revision: Temperature conditions have been given in the manuscript.

  1. The supporting information seems the same with original version.

The authors mention that the added results of cell viability assay in supporting information (Page 7, line 218). But, the results are missing.

 Response and revision:  The file of the supporting information has been resubmitted.

  1. Caption of Figure 5 seems wrong.

This reviewer speculates that the Lane 0 is DNA+Ru(20uM) without light irradiation and Lane 1 is DNA only (without addition of Ru) with light irradiation. 

 Response and revision: The errors have been corrected.

  1. There are typos in text.

For example,

"nicked nicked circularsincreased" (Page 5, line 166)

"than that of than" (Page 6, line 173)

This reviewer recommends careful check of their English.

 Response and revision: The errors have been corrected.

5, Some "1" in "compound 1" are not bolded in paragraph of "2.6 Photocytotoxicity".

 Response and revision: The errors have been corrected.

  1. Line break should be added before "3.3 Photoinduced DNA cleavage".

Response and revision: The error has been corrected.